# The Thrombopoietic Signature of Preeclampsia: Diagnostic and Monitoring Insights from the Immature Platelet Fraction

**DOI:** 10.3390/diagnostics16010044

**Published:** 2025-12-23

**Authors:** Ilkay Er, Senol Sentürk, Medeni Arpa, Nalan Kuruca

**Affiliations:** 1Department of Pediatrics, Division of Neonatology, Faculty of Medicine, Recep Tayyip Erdogan University, Rize 53020, Turkey; 2Department of Obstetrics and Gynecology, Faculty of Medicine, Recep Tayyip Erdogan University, Rize 53020, Turkey; 3Department of Medical Biochemistry, Faculty of Medicine, Recep Tayyip Erdogan University, Rize 53020, Turkey; medeni.arpa@erdogan.edu.tr

**Keywords:** preeclampsia, immature platelet fraction, platelet indices, biomarker, thrombopoiesis, treatment response

## Abstract

**Background:** Preeclampsia is a major obstetric disorder characterized by platelet activation and dysregulated thrombopoiesis. While conventional platelet indices reflect platelet morphology, the immature platelet fraction (IPF) provides insight into thrombopoietic activity. This study assessed IPF discrimination at presentation and its early post-treatment change in preeclampsia while controlling for potential confounding factors. **Methods:** In a prospective design, demographic and laboratory parameters—particularly platelet indices—were evaluated in women with preeclampsia and normotensive pregnant controls. Measurements were obtained at diagnosis and repeated 24–48 h after treatment, including initiation of medical treatment or delivery. Logistic regression and ROC analyses were performed, adjusting for age and gestational age. **Results:** Sixty-four women with preeclampsia and 25 normotensive controls were included; the preeclampsia group was older (31.3 ± 5 vs. 28.4 ± 4 years), and delivery occurred in 73.4%. At diagnosis, IPF, MPV, and PDW were higher, and platelet counts were lower compared with controls. After treatment, IPF decreased markedly (ΔIPF = 3.4; *p* < 0.001), accompanied by reductions in MPV and PDW, while platelet counts remained unchanged in the preeclampsia group. ΔIPF showed subtype-related differences, being higher in late-onset preeclampsia. Only IPF retained an independent association with preeclampsia (OR = 27.29; *p* = 0.006), whereas age, platelet count, MPV, PDW, BUN, and CRP were not significant. On ROC analysis, IPF demonstrated strong diagnostic performance (AUC = 0.992; cut-off ≥4%), with 98.4% sensitivity and 100% specificity. **Conclusions:** Easily measurable as part of a routine complete blood count, IPF may support diagnostic evaluation and clinical monitoring, consistent with its early post-treatment decline and subtype-related patterns.

## 1. Introduction

Preeclampsia remains a leading cause of maternal and perinatal morbidity worldwide. Ongoing research continues to focus on identifying reliable biomarkers to enable earlier diagnosis and improve clinical management [1,2]. Clinically, preeclampsia is defined by the new onset of hypertension after the 20th week of gestation, often accompanied by proteinuria or signs of maternal organ dysfunction [3]. Its pathogenesis is not fully understood; however, mounting evidence suggests that hypertensive disorders of pregnancy are driven by placental dysregulation involving inflammation, endothelial injury, and microthrombus formation—processes in which platelet activation plays a pivotal role and which may offer valuable targets for biomarker development [1,4,5,6].

Since direct measurement of platelet activation is challenging, surrogate indices of platelet size (mean platelet volume, MPV; platelet distribution width, PDW) and production (immature platelet fraction, IPF) have been increasingly utilized as indicators of heightened platelet activity. Among these, immature platelets are of particular interest; being younger, larger, and more thrombogenic than mature platelets, they serve as reliable markers of accelerated platelet turnover and bone marrow activity [6,7,8,9]. Although their precise role in the development of preeclampsia and other hypertensive disorders of pregnancy remains under investigation, accumulating evidence suggests that immature platelets may contribute to the disease pathophysiology [10,11,12,13]. Nevertheless, evidence on the clinical relevance of IPF beyond initial diagnosis, especially its ability to reflect treatment response and disease dynamics, is still limited [9,10,11,13].

In light of these considerations, the present study was designed to comprehensively evaluate the clinical significance of IPF in preeclampsia. Specifically, we aimed to assess its diagnostic discrimination at presentation and its early post-treatment change, while carefully controlling for potential confounding factors.

## 2. Materials and Methods

### 2.1. Study Design and Population

Women in the second and third trimesters of pregnancy who attended the Obstetrics Department of Recep Tayyip Erdogan University between December 2023 and December 2024 were recruited for this prospective observational case–control study. The study protocol was approved by the institutional ethics committee, and written informed consent was obtained from all participants prior to enrollment.

The study population was stratified into two groups: a preeclampsia group, comprising women newly diagnosed according to the American College of Obstetricians and Gynecologists (ACOG) criteria, and a control group of healthy, normotensive pregnant women beyond 20 weeks of gestation. ACOG defines preeclampsia as new-onset hypertension after 20 weeks [systolic blood pressure (SBP) ≥ 140 mmHg or diastolic blood pressure (DBP) ≥ 90 mmHg on two occasions at least four hours apart] in a previously normotensive woman, accompanied by either proteinuria [≥300 mg per 24 h urine collection, urine protein/creatinine (UPC) ratio ≥ 0.3, or dipstick ≥ 1+] or, in the absence of proteinuria, new maternal organ dysfunction [14]. Non-severe preeclampsia is characterized by elevated blood pressure with proteinuria but without evidence of severe features. Severe preeclampsia, on the other hand, is diagnosed when one or more of the following are present: SBP ≥ 160 mmHg or DBP ≥ 110 mmHg (confirmed at least four hours apart), thrombocytopenia (<100,000/mm^3^), impaired liver function (elevated AST or ALT levels) with or without right upper quadrant or epigastric pain, progressive renal insufficiency (serum creatinine > 1.1 mg/dL or doubling of baseline), pulmonary edema, or new-onset cerebral or visual disturbances [14]. Management is tailored to disease severity and gestational age: close maternal and fetal monitoring with antihypertensive therapy and corticosteroids for fetal lung maturation in non-severe cases before term, and timely delivery as the only definitive treatment, particularly in severe cases or at ≥37 weeks of gestation. Magnesium sulfate is recommended for seizure prophylaxis in severe preeclampsia, while supportive care plays a central role in optimizing maternal and perinatal outcomes [14,15]. Independent of severity, preeclampsia is also classified according to gestational age at onset as early-onset (<34 + 0 weeks) or late-onset (≥34 + 0 weeks), reflecting its biological heterogeneity, with early-onset disease primarily related to impaired placentation and late-onset preeclampsia more closely associated with maternal cardiovascular, metabolic, and inflammatory factors [14].

Gestational hypertension refers to new-onset hypertension—defined as SBP ≥ 140 mmHg or DBP ≥ 90 mmHg on two separate measurements at least 4 h apart—that develops after 20 weeks of gestation without accompanying proteinuria or other features of preeclampsia. HELLP syndrome is a severe, potentially life-threatening form of preeclampsia characterized by hemolysis, elevated liver enzyme levels, and thrombocytopenia [14].

To ensure that the study population consisted exclusively of women with preeclampsia and healthy controls from otherwise uncomplicated pregnancies, strict selection criteria were applied. Women with chronic hypertension predating pregnancy, hypertension diagnosed before 20 weeks of gestation, gestational hypertension, or HELLP syndrome were excluded. Conditions that could influence maternal hematological or obstetric outcomes—including pre-pregnancy or early-pregnancy obesity (BMI ≥ 30 kg/m^2^), multiple or molar pregnancies, a history of spontaneous abortion before 20 weeks, pregestational or gestational chronic diseases (e.g., diabetes mellitus, thyroid disorders, rheumatologic, renal, vascular, or autoimmune conditions), maternal infections, and pregnancies complicated by chromosomal abnormalities—were not eligible for participation. Pregnant with thrombocytopenia due to causes other than preeclampsia, as well as exposure to factors known to affect platelet indices—such as medications (e.g., antithrombotic or hematologic agents), smoking, or alcohol and substance use—were not included in the study [14,16,17]. Finally, refusal to provide informed consent or loss to follow-up was also considered among the criteria for exclusion.

### 2.2. Blood Sampling and Analysis

In patients with newly diagnosed preeclampsia, venous blood samples were obtained twice: at diagnosis and after routine treatment, including antihypertensive therapy, magnesium sulfate when indicated, antenatal corticosteroids if required, and/or delivery. Blood was collected simultaneously into EDTA tubes for complete blood count (CBC) and IPF analysis, and into plain serum tubes for liver and renal function tests (aspartate aminotransferase [AST], alanine aminotransferase [ALT], serum creatinine, and blood urea nitrogen [BUN]) as well as C-reactive protein (CRP). CRP was included to evaluate the systemic inflammatory response associated with preeclampsia. However, post-treatment CRP measurements were not included in the study, as they could be affected by peripartum factors such as surgical stress or anesthesia. Hematological analyses from EDTA tubes included standard CBC parameters such as white blood cell (WBC) count, hemoglobin (Hb), and hematocrit (Hct), as well as platelet (PLT) indices such as MPV and PDW, measured using a multi-parameter automated hematology analyzer.

Considering the biological half-lives of AST (~17 ± 5 h), ALT (~47 ± 10 h), serum creatinine (~4 h), and blood urea nitrogen (BUN) (~4–6 h), a 24–48 h (mean interval: 36 h) was chosen to allow sufficient time for treatment-related changes in these biochemical parameters to occur in the preeclampsia group [18]. Venous blood samples collected at the time of diagnosis and 24–48 h after treatment were designated as sample 1 and sample 2, respectively (e.g., IPF-1 and IPF-2). The difference between these two measurements was calculated and reported as the delta (Δ) value, reflecting short-term post-treatment change. In the control group, a single baseline venous blood sample was obtained after the 20th week of gestation and analyzed following the same procedures. All samples with the UPC ratio were collected as part of routine clinical care, without any additional invasive interventions.

### 2.3. Measurement of IPF

All CBC parameters, including IPF, were obtained from the same EDTA sample and analyzed on the same hematology analyzer to ensure consistency of results and minimize preanalytical variability. Specifically, venous blood samples were processed using the Mindray BC-6000 automated hematology analyzer (Mindray Bio-Medical Electronics Co., Ltd., Shenzhen, China), which is based on flow cytometry technology. The instrument applies a fluorescent dye that binds to platelet RNA, enabling the identification of immature platelets characterized by larger size and higher fluorescence intensity. Scatter plots were generated to differentiate platelet subpopulations according to cell volume and RNA content, and IPF was determined as the percentage of reticulated (immature) platelets within the total platelet count [19].

### 2.4. Statistical Analyses

All statistical analyses were performed using IBM SPSS Statistics version 29.0 (IBM Corp., Armonk, NY, USA) and MedCalc Statistical Software version 19.1.3 (MedCalc Software bv, Ostend, Belgium; https://www.medcalc.org, accessed on 10 September 2025). The normality of distribution for continuous variables was evaluated with the Kolmogorov–Smirnov test. Data are summarized as mean ± standard deviation (SD), median (minimum–maximum), or frequency and percentage, as appropriate. Group comparisons were conducted using Student’s *t*-test or the Mann–Whitney U test, depending on the distribution of variables, while categorical data were analyzed using the Chi-square test or Fisher’s exact test. For paired measurements within groups, the paired-sample *t*-test or the Wilcoxon signed-rank test was applied. Diagnostic performance of hematological parameters was examined by receiver operating characteristic (ROC) curve analysis, and the area under the curve (AUC), sensitivity, specificity, and optimal cut-off values were reported. Cut-off thresholds were determined using the maximum Youden index (J = sensitivity + specificity − 1). Correlations between continuous variables were assessed using Spearman’s rank correlation coefficient. Logistic regression analysis was performed to identify independent predictors of preeclampsia. A *p*-value < 0.05 was considered statistically significant.

## 3. Results

### 3.1. Study Enrollment and Characteristics of the Study Population

During the study period, 1082 pregnant women presented to the obstetrics outpatient clinic of our tertiary referral hospital. After applying the exclusion criteria, eligible participants were included in the final analysis. Among them, 64 women (5.9%) were newly diagnosed with preeclampsia and were designated as the preeclampsia group, while 25 rigorously selected healthy pregnant women were designated as the control group. The high exclusion rate reflects the referral profile of our center, which routinely manages complex maternal and obstetric cases (Figure 1).

Baseline maternal and neonatal characteristics of the study population are summarized in Table 1. Pregnant women with preeclampsia were significantly older than pregnant controls and delivered at earlier gestational ages (*p* = 0.008 and *p* < 0.001, respectively). Parity was lower in the preeclampsia group (*p* = 0.005), and the rate of cesarean delivery was substantially higher (92.2% vs. 64%, *p* = 0.002). Both SBP and DBP were markedly elevated in women with preeclampsia compared to normotensive controls (both *p* < 0.001), reflecting the hemodynamic profile of the disease. Neonates born to women with preeclampsia had significantly lower birth weights (*p* < 0.001), whereas Apgar scores at 1 and 5 min were comparable between groups.

Gestational age in the preeclampsia group was recorded at the time of diagnosis. Since the majority of patients (approximately 96.8%) either underwent delivery soon after diagnosis or following appropriate medical intervention (73.4%), or showed a rapid clinical response to treatment (23.4%), the recorded value was almost identical to the gestational age at delivery or treatment completion. For the remaining minority, pregnancy was prolonged for a short period under expectant management until maternal stabilization or fetal maturity was achieved, which did not significantly influence the overall gestational age profile.

### 3.2. Comparison of Initial Laboratory Profiles Between Groups

Initial laboratory findings, adjusted for age and gestational week, are presented in Table 2. Compared to healthy pregnant controls, pregnant women with preeclampsia demonstrated marked hematologic differences, predominantly characterized by a significant elevation in the IPF (8.0 ± 4.8 vs. 2.4 ± 0.9%; *p* < 0.001). This marked elevation in IPF-1 was accompanied by significantly higher MPV-1 and PDW-1 values (MPV-1: 12.3 ± 1.5 vs. 10.8 ± 1.1 fL, *p* = 0.004; PDW-1: 16.3 vs. 16.2%, *p* = 0.033) and a notable reduction in PLT-1 (219.5 vs. 275 × 10^3^/μL; *p* < 0.001). The UPC ratio was above the normal reference range in the preeclampsia group, consistent with the characteristic renal involvement of the disorder. No significant differences were observed between the groups in WBC-1, Hb-1, Hct-1, BUN-1, creatinine-1, ALT-1, AST-1, or CRP-1 levels (*p* > 0.05).

### 3.3. Treatment-Related Changes in Laboratory Parameters Among Pregnant Women with Preeclampsia

Table 3 illustrates the changes in laboratory parameters before and after treatment, together with the corresponding Δ values, in pregnant women with preeclampsia. After treatment, IPF decreased significantly (8.0 ± 4.8 to 4.58; ΔIPF = 3.4 ± 1.7; *p* < 0.001). MPV also decreased (12.3 ± 1.5 to 11.5 ± 1.3; ΔMPV = 0.83 ± 0.86; *p* < 0.001), accompanied by a modest reduction in PDW (*p* = 0.028). Hemoglobin and hematocrit likewise showed significant decreases (*p* = 0.002 and *p* = 0.003, respectively), changes that may occur as part of the expected post-treatment course, whereas PLT and WBC counts remained unchanged (*p* = 0.561 and *p* = 0.979). The UPC ratio also decreased significantly (5.1 ± 1.1 to 1.2 ± 0.7; *p* = 0.005). Liver and renal function tests showed no significant changes, except for a small increase in ALT (*p* = 0.046), while creatinine did not reach statistical significance (*p* = 0.065).

### 3.4. Severity-Related Differences in Δ Laboratory Parameters in the Preeclampsia Group

Table 4 presents the variation in Δ values of laboratory parameters across different levels of preeclampsia severity. ΔIPF was higher in severe preeclampsia compared with non-severe cases (4.4 ± 2.7 vs. 3.2 ± 1.2), although this difference did not reach statistical significance (*p* = 0.108). In contrast, ΔMPV was significantly greater in severe preeclampsia (1.34 ± 0.92 vs. 0.69 ± 0.80, *p* = 0.012). ΔHct was also significantly higher (*p* = 0.029), while the ΔUPC ratio was significantly lower in severe preeclampsia compared to non-severe cases (0.9 ± 0.8 vs. 2.9 ± 1.1; *p* = 0.046), both reflecting more pronounced alterations associated with advanced disease severity. No significant differences were observed in the other Δ values, including biochemical indices.

### 3.5. Gestational Age-Based Differences in Δ Laboratory Parameters in Preeclampsia Group

Regardless of disease severity, comparative analysis of Δ values revealed distinct hematologic response patterns between early-onset and late-onset preeclampsia within the preeclampsia group (Table 5). ΔPLT and ΔMPV were significantly higher in the early onset group (*p* = 0.032 and *p* = 0.034, respectively), whereas ΔIPF was significantly greater in the late-onset group (*p* = 0.038). In addition, ΔALT differed significantly between groups (*p* = 0.031). No significant intergroup differences were observed for the remaining Δ values, including the ΔUPC ratio.

### 3.6. Logistic Regression Analysis of Parameters Associated with Preeclampsia

In the study, univariate logistic regression identified age (*p* = 0.011; OR 1.156, 95% CI 1.034–1.293), PLT count (*p* = 0.008; OR 1.00, 95% CI 1.00–1.00), IPF (OR 31.88, 95% CI 3.93–258.56; *p* = 0.001), MPV (OR 2.50, 95% CI 1.54–4.05; *p* < 0.001), PDW (OR 4.77, 95% CI 1.01–22.56; *p* = 0.049), BUN (OR 1.20, 95% CI 1.07–1.36; *p* = 0.003), and CRP (OR 1.13, 95% CI 1.01–1.26; *p* = 0.030) as significant correlates. In the multivariate model, however, only IPF remained independently associated with preeclampsia (OR 27.29, 95% CI 2.60–86.46; *p* = 0.006), while other variables did not retain statistical significance (Table 6).

### 3.7. Diagnostic Performance of Platelet Indices According to ROC Analysis

ROC analysis demonstrated that IPF-1 had the highest diagnostic accuracy among the evaluated platelet indices for preeclampsia (AUC = 0.992, 95% CI: 0.975–1.008, *p* < 0.001), with a sensitivity of 98.4% and specificity of 100% at a cut-off value of ≥4%. MPV-1 also showed moderate diagnostic performance, with an AUC of 0.775 (95% CI: 0.670–0.880, *p* < 0.001) and optimal discrimination at a threshold of ≥11.1 fL (sensitivity 84.4%, specificity 64%). PLT-1 yielded an AUC of 0.697 (95% CI: 0.587–0.806, *p* < 0.001) with a cut-off value of ≤216.5 × 10^9^/L (sensitivity 48.4%, specificity 92%). PDW-1 demonstrated the lowest diagnostic performance (AUC = 0.625, 95% CI: 0.501–0.749, *p* = 0.047), with 46.9% sensitivity and 80% specificity (Figure 2, Table 7).

Among all platelet indices, IPF exhibited the highest discriminative ability, outperforming MPV, PDW, and platelet count in differentiating preeclampsia from normotensive pregnancies.

## 4. Discussion

In this prospective study, IPF effectively distinguished preeclampsia from normotensive pregnancies and showed superior discriminative performance compared with conventional platelet indices. Moreover, the significant early post-treatment decline in IPF—despite stable platelet counts—was indicative of short-term hematologic changes following standard management. Subtype-related differences between early and late-onset preeclampsia further support its potential role in clinical assessment and monitoring.

Preeclampsia affects approximately 5% (range: 2–15%) of all pregnancies [20,21], a rate comparable to that observed in our cohort (5.9%). The disorder is characterized by a multifactorial pathophysiology involving abnormal placentation, endothelial dysfunction, and an exaggerated inflammatory response. Platelet activation plays a pivotal role in these mechanisms, promoting a proinflammatory and procoagulant milieu that contributes to microthrombus formation and placental hypoxia [1,3,4,5,6]. In our cohort, affected women were generally older, delivered earlier, and had neonates with lower birth weights. Accompanying biochemical and hematologic abnormalities—such as elevated UPC ratios and altered platelet indices—further reflected the multisystemic nature of the disease and its well-recognized maternal and perinatal consequences [14,21].

In response to accelerated platelet consumption, thrombopoietin-mediated compensatory thrombopoiesis promotes the release of immature platelets, resulting in elevated IPF values that reflect enhanced megakaryocytic activity and altered platelet kinetics [5,6,12,22,23]. Previous reports have shown that mean IPF levels during healthy pregnancy typically range between 3% and 5% and rarely exceed 7%, representing a physiologic adaptation of hematopoiesis to gestation [24,25]. Elevated IPF levels in preeclampsia, however, indicate excessive platelet turnover and impaired hemostatic regulation, once other causes of thrombocytopenia are excluded [12,25].

Recent studies have consistently reported elevated IPF values in pregnant women with preeclampsia compared with normotensive pregnant controls. Moraes et al. [10] documented median IPF levels of 8.6% versus 3.8%, Bernstein et al. [11] reported 7.6% versus 4.1%, and Dionisio et al. [9] observed 11% versus 3.9%, all demonstrating significant differences. Consistent with these findings, IPF values in our preeclampsia group were markedly higher than in controls (8 ± 4.8% vs. 2.4 ± 0.9%; *p* < 0.001). The relatively lower IPF values in our control group likely stem from methodological factors—particularly the stricter exclusion criteria—which minimized potential confounding and enabled a more accurate assessment of disease-specific changes.

Consistent with the elevation in IPF, both MPV and PDW values were higher in pregnant women with preeclampsia, whereas platelet counts were lower, reflecting intensified platelet turnover [6,7]. These findings are in line with those reported by Moraes et al. [10] and Dionisio et al. [9], reinforcing the notion that larger, newly released platelets constitute a hematologic hallmark of preeclampsia, despite occasional inconsistencies described in the literature [23].

Our study demonstrated clear hematologic improvements following treatment for preeclampsia. The most striking finding was a significant reduction in IPF (8.0 ± 4.8 to 4.58; ΔIPF = 3.4 ± 1.7; *p* < 0.001), indicating rapid normalization of platelet activity despite stable platelet counts. Concurrent decreases in MPV and PDW support the interpretation that early changes in platelet indices may better reflect hematologic recovery than static measures. Expected reductions in Hb, Hct, and UPC ratio, together with a mild ALT elevation—likely reflecting transient hepatic perfusion changes—were also observed [14]. Accordingly, post-treatment changes in IPF were examined to characterize short-term hematologic responses following routine management and to clarify whether such changes could contribute to early clinical decision-making.

Comparative analysis of Δ values showed that hematologic and biochemical changes became more pronounced with increasing preeclampsia severity. Although ΔIPF tended to be higher in severe cases, the difference was not statistically significant, consistent with Bernstein et al. [11], who observed similar findings between preeclampsia and HELLP syndrome—likely reflecting limited sample size rather than a true absence of association. Conversely, Aggarwal et al. [26] demonstrated a progressive rise in IPF with disease severity, from 10.1% in gestational hypertension to 18.1% in eclampsia (*p* < 0.001). However, in our cohort, ΔMPV also increased significantly with disease severity. Prior studies showing elevated MPV in gestational hypertension likewise suggest that this alteration may precede the clinical onset of preeclampsia [27]. Differences in study design, population, and methodology may explain discrepancies across studies but collectively underscore the value of dynamic platelet indices—particularly IPF and MPV—in understanding disease pathophysiology.

Beyond severity-based comparisons, early-onset preeclampsia showed greater increases in ΔPLT and ΔMPV, whereas ΔIPF was significantly higher in late-onset preeclampsia (median 3.4 [0.6–9.1] vs. 2.4 [1.5–7.9]; *p* = 0.038) [1,20]. This pattern may suggest an additional hematologic aspect of maternal involvement in late-onset disease and warrants confirmation in larger cohorts.

To further evaluate the diagnostic performance of platelet indices, we conducted logistic regression analyses, which showed that IPF was the only variable independently associated with preeclampsia, whereas age, platelet count, MPV, PDW, BUN, and CRP lost significance in the multivariate model. This finding underscores the disease-specific diagnostic association of IPF, reflecting increased platelet production and turnover characteristic of preeclampsia. The markedly elevated odds ratio (OR = 27.29, 95% CI: 2.60–86.46; *p* = 0.006) observed in our cohort further supports the strength of this association, compared with conventional indices that may be more susceptible to nonspecific inflammatory or hemodilutional influences.

Moreover, IPF demonstrated the highest diagnostic accuracy among platelet parameters in distinguishing pregnant women with preeclampsia from normotensive pregnant controls in our cohort (AUC = 0.992; cut-off ≥4%; sensitivity 98.4%, specificity 100%). MPV also showed good diagnostic accuracy (AUC = 0.775), whereas platelet count and PDW exhibited only modest performance (AUC = 0.697 and 0.625, respectively). Comparable findings have been reported in studies evaluating IPF in hypertensive disorders of pregnancy. For instance, Hayuningsih et al. [28] observed significantly higher IPF values in pregnant women with preeclampsia than in normotensive controls, with IPF demonstrating the best diagnostic performance among platelet indices (AUC = 0.88; 95% CI: 0.78–0.95; cut-off >6.5%). Similarly, Moraes et al. [10] reported an AUC of 0.83 for IPF in differentiating hypertensive disorders of pregnancy from controls, while Aggarwal et al. [26] further demonstrated strong discriminatory ability (AUC = 0.879; cut-off >13.3%) in distinguishing severe preeclampsia and eclampsia from milder hypertensive forms. Notably, both studies assessed IPF as a standalone hematologic parameter, which limits direct comparison with other platelet indices evaluated in our study.

Conversely, Dionisio et al. [9], in a recent study evaluating platelet indices in preeclampsia, identified MPV as the best-performing parameter (AUC = 0.833), whereas IPF% showed only moderate diagnostic accuracy (AUC = 0.801; cut-off = 7.8%; sensitivity 70%, specificity 92%). In the same cohort, the platelet large cell ratio (PLCR)—defined as the proportion of platelets exceeding 12 fL in volume—demonstrated a diagnostic performance comparable to that of IPF (AUC = 0.814; sensitivity 70%, specificity 92%). Similarly, Tesfay et al. [29] and Walle et al. [30] also reported MPV as the most informative platelet index in preeclampsia. Other investigations, including those by Thalor et al. [31], who emphasized PDW, and Umezuluike et al. [32], who focused on plateletcrit and P-LCR, likewise revealed significant differences in platelet indices between normotensive and preeclamptic pregnancies. However, none of these studies assessed IPF, which may provide a more direct measure of thrombopoietic activity than morphology-based platelet parameters.

In our cohort, IPF clearly outperformed MPV and other platelet indices, emerging as a more robust and physiologically relevant marker of platelet dynamics in preeclampsia. This divergence likely reflects both methodological and biological factors. Specifically, our study applied strict exclusion criteria and assessed IPF at the time of diagnosis, enabling the detection of earlier and subtler changes in platelet activation and production. Furthermore, the distinct biological basis of IPF—quantifying RNA-positive immature platelets that represent true thrombopoietic activity rather than morphology-based platelet enlargement measured by MPV, PDW, or PLCR—may further explain its superior diagnostic performance [33,34]. Considering the limitations of current screening biomarkers such as Doppler indices and angiogenic factors [4,35,36,37], which are often costly and technically demanding, IPF offers a simple, inexpensive, and readily accessible alternative, measurable as part of a routine complete blood count through standard hematology analyzers, with promising potential for future clinical integration.

### Strengths and Limitations

In this study, although preeclampsia diagnosis was established according to ACOG criteria, IPF was evaluated among hematologic markers to provide physiological insight beyond clinical confirmation. It demonstrated enhanced discriminative performance together with onset-related heterogeneity and treatment-associated hematologic changes independent of platelet counts—features that have not been well characterized in previous studies.

These findings should be interpreted in the context of several limitations. The relatively modest sample size—particularly within the severity-based subgroups—may have limited the statistical power to detect subtle differences, reflecting the inherent challenges of recruiting women with severe preeclampsia who meet strict inclusion criteria. Because our institution is a tertiary referral center, the selection of suitable controls was further constrained, as many referred pregnancies presented with comorbid medical or obstetric conditions that precluded participation. The comparatively lower IPF values observed in the control group likely stem from methodological factors—particularly the stringent exclusion criteria—which minimized potential confounders of platelet turnover and enabled a more accurate assessment of disease-specific changes. Importantly, women with chronic hypertension, routine aspirin use, and other comorbidities known to affect platelet indices were deliberately excluded to reduce bias. In addition, the single-center design may restrict the external generalizability of our findings, although it ensured methodological uniformity and consistency in sample processing. Nonetheless, the use of a well-defined cohort and a standardized analytical protocol strengthens the internal validity and robustness of the results.

## 5. Conclusions

This study demonstrates that IPF shows strong discriminative performance in preeclampsia, with diagnostic performance exceeding that of conventional platelet indices—such as MPV—that have traditionally dominated the literature. Beyond its ability to distinguish clinically established preeclampsia from normotensive pregnancies at presentation, IPF showed a significant early post-treatment decline and distinct onset-related patterns, supporting its potential role in subtype-sensitive clinical assessment. Modest subtype-specific differences in liver enzymes further supported the biological heterogeneity of preeclampsia. Easily measurable as part of routine complete blood count analysis, IPF may support short-term clinical monitoring of preeclampsia. Future multicenter studies with larger cohorts are warranted to validate these findings and to further clarify the clinical role of IPF in preeclampsia.

## Figures and Tables

**Figure 1 diagnostics-16-00044-f001:**
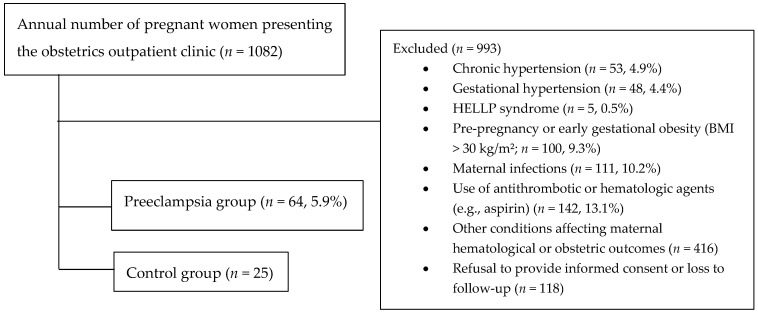
Flowchart of the enrollment, exclusion criteria, and classification of pregnant women into the preeclampsia and control groups.

**Figure 2 diagnostics-16-00044-f002:**
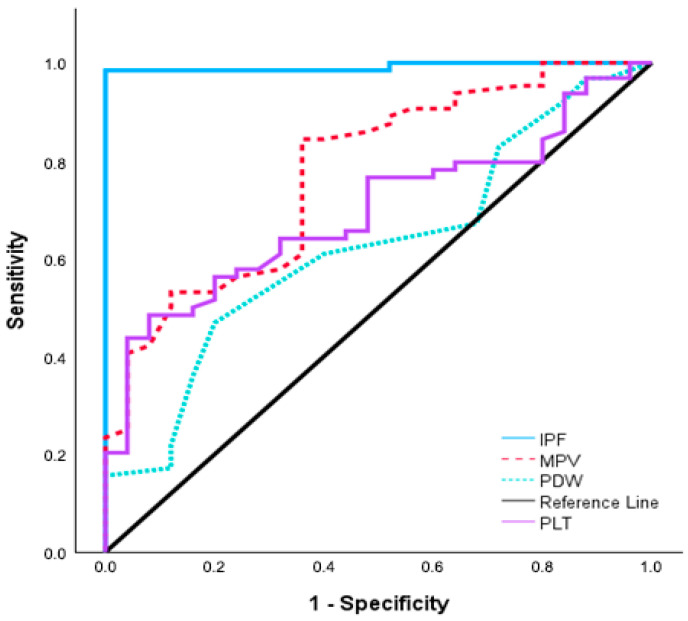
ROC curves illustrating the diagnostic performance of PLT, MPV, PDW, and IPF in preeclampsia.

**Table 1 diagnostics-16-00044-t001:** Maternal and neonatal characteristics of the preeclampsia and control groups.

Characteristics	Preeclampsia Group *n* = 64 ^1^	Control Group *n* = 25 ^1^	*p*-Value
**Age**, years	31.3 ± 5.0	28.4 ± 4.0	**0.008 ^&^**
**Gestational weeks**	35.2 (26.2–39.6)	37.4 (34–40.5)	**<0.001 ***
**Gravidity**	1 (1–7)	2 (1–4)	0.561 *
**Parity**	0 (0–4)	1 (0–3)	**0.005 ***
**Nulliparous**	36 (56.3)	11(44)	0.298 *
**Previous abortions**	14 (32)	8 (21.9)	0.320 *
**SBP**, mmHg	147.5 (130–180)	110 (100–120)	**<0.001 ***
**DBP**, mmHg	90 (80–117)	70 (60–80)	**<0.001 ***
**Cesarean delivery**	59 (92.2)	16 (64)	**0.002 ***
**Birth weight**, kg	2442 ± 799	2944 ± 419	**<0.001 ^&^**
**Apgar score**, 1st min	8 (3–9)	8 (7–9)	0.267 *
**Apgar score**, 5th min	9 (6–10)	9 (8–10)	0.221 *

SBP: systolic blood pressure; DBP: diastolic blood pressure; ^1^: number (percentage), mean ± SD, median (min–max); bold *p*-values indicate statistical significance (*p* < 0.05); *: Mann–Whitney U test; ^&^: Student’s *t*-test.

**Table 2 diagnostics-16-00044-t002:** Comparison of initial hematological and biochemical parameters between the preeclampsia and control groups.

Parameters ^#^	Preeclampsia Group *n* = 64 ^1^	Control Group *n* = 25 ^1^	*p*-Value
**WBC-1** (×10^3^/μL)	10.0 (5.8–19.5)	10.7 (5.1–17.7)	0.519 *
**Hb-1** (g/dL)	11.4 ± 1.2	11.4 ± 1.2	0.759 ^&^
**Hct-1** (%)	34.3 ± 3.5	34.5 ± 3.6	0.829 ^&^
**PLT-1** (×10^3^/μL)	219.5 (120–402)	275 (176–444)	**<0.001 ***
**MPV-1** (fL)	12.3 ± 1.5	10.8 ± 1.1	**0.004** ^&^
**PDW-1** (%)	16.3 (15.6–17.4)	16.2 (15.6–16.7)	**0.033 ***
**IPF-1** (%)	8.0 ± 4.8	2.4 ± 0.9	**<0.001** ^&^
**BUN-1** (mg/dL)	19 (9–38)	16 (10–24)	0.069 *
**Creatinine-1** (mg/dL)	0.6 ± 0.5	0.5 ± 0.3	0.066 ^&^
**ALT-1** (U/L)	11 (5–128)	9 (5–18)	0.140 *
**AST-1** (U/L)	17 (5–163)	15 (10–21)	0.110 *
**CRP-1** (mg/L)	8.2 (0–28.8)	4.6 (1.4–12.9)	0.060 *
**UPC ratio-1**	5.1 ± 1.1	-	-

^#^: Adjusted for age and birth week. WBC: white blood cell; Hb: hemoglobin; Hct: hematocrit; PLT: platelet; MPV: mean platelet volume; PDW: platelet distribution width; IPF: immature platelet fraction; BUN: blood urea nitrogen; ALT: alanine aminotransferase; AST: aspartate aminotransferase; CRP: C-reactive protein; UPC ratio: urine protein/creatinine ratio; ^1^: number (percentage), mean ± SD, median (min–max); bold *p*-values indicate statistical significance (*p* < 0.05); *: Mann–Whitney U test; ^&^: Student’s *t*-test.

**Table 3 diagnostics-16-00044-t003:** Changes in laboratory parameters before and after treatment in pregnant women with preeclampsia.

Parameters	Preeclampsia Group *n* = 64 ^1^	*p*-Value
**WBC-1**	10.0 (5.8–19.5)	0.979 ^¥^
**WBC-2**	10.6 (4.0–19.3)	
**ΔWBC**	−0.27 (−0.21–1.74)	
**Hb-1**	11.2 (8.9–14.5)	**0.002 ^¥^**
**Hb-2**	11 (8–13.3)	
**ΔHb**	0.3 (0.2–1.3)	
**Hct-1**	34.2 (27.9–44.9)	**0.003 ^¥^**
**Hct-2**	33 (22.6–81.9)	
**ΔHct**	1.2 (1.3–37)	
**PLT-1**	219.5 (120.0–402.0)	0.561 ^¥^
**PLT-2**	223.0 (119.0–395.0)	
**ΔPLT**	−5.800 (−1.350–7.000)	
**MPV-1**	12.3 ± 1.5	**<0.001 ^§^**
**MPV-2**	11.5 ± 1.3	
**ΔMPV**	0.83 ± 0.86	
**PDW-1**	16.3 (15.6–17.4)	**0.028** ^¥^
**PDW-2**	16.2 (15.3–17.1)	
**ΔPDW**	1.2 (0.3–2.1)	
**IPF-1**	8.0 ± 4.8	**<0.001 ^§^**
**IPF-2**	4.58	
**ΔIPF**	3.4 ± 1.7	
**BUN-1**	20 ± 7	0.566 ^§^
**BUN-2**	21 ± 9	
**ΔBUN**	−0.5 ± 0.7	
**Creatinine-1**	0.6 ± 0.5	0.065 ^§^
**Creatinine-2**	0.8 ± 0.4	
**Δ Creatinine**	−0.2 ± 0.5	
**ALT-1**	10.5 (5–128)	**0.046 ^¥^**
**ALT-2**	12 (5–50)	
**ΔALT**	−1.9 (−0.3–70)	
**AST-1**	16.5 (5–163)	0.214 ^¥^
**AST-2**	19 (5–100)	
**ΔAST**	−2.9 (−1.9–65)	
**UPC Ratio-1**	5.1 ± 1.1	**0.005 ^§^**
**UPC Ratio-2**	1.2 ± 0.7	
**ΔUPC Ratio**	1.7 ± 0.4	

WBC: white blood cell; Hb: hemoglobin; Hct: hematocrit; PLT: platelet; MPV: mean platelet volume; PDW: platelet distribution width; IPF: immature platelet fraction; BUN: blood urea nitrogen; ALT: alanine aminotransferase; AST: aspartate aminotransferase; UPC ratio: urine protein/creatinine ratio; ^1^: mean ± SD, median (min–max); bold *p*-values indicate statistical significance (*p* < 0.05); *p*^§/¥^: the differences of in-group Δ values analyzed with paired-samples *t*-test § or Wilcoxon signed rank test ¥.

**Table 4 diagnostics-16-00044-t004:** Variation in Δ values of laboratory parameters according to severity in the preeclampsia group.

Parameters	Non-Severe Preeclampsia Group, *n* = 50 ^1^	Severe Preeclampsia Group, *n* = 14 ^1^	*p*-Value
**ΔWBC**	−15 (−11.3–8.41)	−535 (−8.2–6.81)	0.884 ^¥^
**ΔHb**	0.1 (−1.6–2.4)	0.7 (−0.6–3.1)	0.095 ^¥^
**ΔHct**	0.25 (−47.9–7.1)	1.5 (−1.7–9.5)	**0.029 ^¥^**
**ΔPLT**	−4.5 (−82.7–105)	7.5 (−27–87)	0.142 ^¥^
**ΔMPV**	0.69 ± 0.8	1.34 ± 0.92	**0.012 ^§^**
**ΔPDW**	0.1 (−0.9–1.3)	0.2 (−0.6–1.1)	0.630 ^¥^
**ΔIPF**	3.2 ± 1.2	4.4 ± 2.7	0.108 ^§^
**ΔBUN**	−0.3 ± 5.6	−1.3 ± 12.0	0.774 ^§^
**ΔCreatinine**	−0.02 (−0.45–0.44)	0.06 (−1.41–0.13)	0.196 ^¥^
**ΔALT**	−0.5 (−32–6)	0 (−16–82)	0.344 ^¥^
**ΔAST**	−1.5 (−55–19)	0.5 (−81–123)	0.105 ^¥^
**ΔUPC Ratio**	2.9 ± 1.1	0.9 ± 0.8	**0.046 ^§^**

WBC: white blood cell; Hb: hemoglobin; Hct: hematocrit; PLT: platelet; MPV: mean platelet volume; PDW: platelet distribution width; IPF: immature platelet fraction; BUN: blood urea nitrogen; ALT: alanine aminotransferase; AST: aspartate aminotransferase; UPC ratio: urine protein/creatinine ratio; ^1^: mean ± SD, median (min–max); bold *p*-values indicate statistical significance (*p* < 0.05); *p*^§/¥^: the differences of in-group Δ values analyzed with paired-samples *t*-test § or Wilcoxon signed rank test ¥.

**Table 5 diagnostics-16-00044-t005:** Comparison of Δ laboratory parameters between early-onset and late-onset preeclampsia in the preeclampsia group.

Parameters	Early Onset Preeclampsia Group, *n* = 21 ^1^	Late-Onset Preeclampsia Group, *n* = 43 ^1^	*p*-Value
**ΔWBC**	0.46 (−6.9–8.41)	−0.39 (−11.3–7.02)	0.637 ^¥^
**ΔHb**	0.2 (−0.3–3.1)	0.1 (−1.6–2.4)	0.215 ^¥^
**ΔHct**	1.1 (−1.1–9.5)	0.3 (−47.9–9)	0.211 ^¥^
**ΔPLT**	224.81 ± 430.76	−386 ± 240.24	**0.032 ^§^**
**ΔMPV**	1.16 ± 0.94	0.67 ± 0.79	**0.034 ^§^**
**ΔPDW**	0 (−0.9–0.6)	0.1 (−0.4–1.3)	0.106 ^¥^
**ΔIPF**	2.4 (1.5–7.9)	3.4 (0.6–9.1)	**0.038 ^¥^**
**ΔBUN**	−1.52 ± 10.62	−0.05 ± 5.17	0.552 ^§^
**ΔCreatinine**	−0.04 (−1.41–0.29)	−0.01 (−0.22–0.44)	0.520 ^¥^
**ΔALT**	−1 (−16–2)	1 (−32–82)	**0.031 ^¥^**
**ΔAST**	−1 (−81–5)	−1 (−55–123)	0.150 ^¥^
**ΔUPC Ratio**	1.7 ± 1.3	1.9 ± 0.8	0.785 ^§^

WBC: white blood cell; Hb: hemoglobin; Hct: hematocrit; PLT: platelet; MPV: mean platelet volume; PDW: platelet distribution width; IPF: immature platelet fraction; BUN: blood urea nitrogen; ALT: alanine aminotransferase; AST: aspartate aminotransferase; UPC ratio: urine protein/creatinine ratio; ^1^: mean ± SD, median (min–max); bold *p*-values indicate statistical significance (*p* < 0.05); *p*^§/¥^: the differences of in-group Δ values analyzed with paired-sample *t*-test § or Wilcoxon signed rank test ¥.

**Table 6 diagnostics-16-00044-t006:** The multivariate logistic regression of the parameters for preeclampsia.

Variables	OR	95% CI		*p*-Value
		Lower	Upper	
**Age**	1.094	0.799	1.499	0.575
**PLT-1**	1.000	1.000	1.000	0.974
**MPV-1**	1.019	0.944	1.1	0.625
**PDW-1**	0.973	0.657	1.441	0.890
**IPF-1**	27.291	2.6	86.46	**0.006**
**BUN-1**	1.083	0.815	1.438	0.583
**CRP-1**	1.123	0.751	1.677	0.573

OR: odds ratio; CI: confidence interval; PLT: platelet; IPF: immature platelet fraction; MPV: mean platelet volume; PDW: platelet distribution width; BUN: blood urea nitrogen; CRP: C-reactive protein; bold *p*-value indicates statistical significance (*p* < 0.05).

**Table 7 diagnostics-16-00044-t007:** Diagnostic performance of PLT, MPV, PDW, and IPF based on ROC analysis for preeclampsia.

Parameters	Cut-Off	AUC (95% CI)	*p*-Value	Sensitivity (%)	Specificity (%)
**PLT-1**	≤216.5	0.697 (0.587–0.806)	**<0.001**	48.4%	92%
**MPV-1**	≥11.1	0.775 (0.670–0.880)	**<0.001**	84.4%	64%
**PDW-1**	≥16.4	0.625 (0.501–0.749)	**0.047**	46.9%	80%
**IPF-1**	≥4	0.992 (0.975–1.008)	**<0.001**	98.4%	100%

PLT: platelet; MPV: mean platelet volume; PDW: platelet distribution width; IPF: immature platelet fraction; AUC: area under the curve; CI: confidence interval; bold *p*-values indicate statistical significance (*p* < 0.05).

## Data Availability

The original contributions presented in this study are included in the article. Further inquiries can be directed to the corresponding author.

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
