# Peer review of "The Thrombopoietic Signature of Preeclampsia: Diagnostic and Monitoring Insights from the Immature Platelet Fraction"

_diagnostics, 2025, doi:10.3390/diagnostics16010044_

Round 1

Reviewer 1 Report

Comments and Suggestions for Authors

The paper reports an interesting study evaluating the diagnostic value of the immature platelet fraction (IPF) and its post-treatment changes in preeclampsia. This small prospective study, involving 64 women with preeclampsia and 25 healthy normotensive pregnant controls, highlights the potential of IPF as a dynamic marker of disease activity. These findings may prove useful given the complexity of preeclampsia diagnosis. IPF measurement is readily accessible in most laboratories, making it a simple and valuable tool for the prevention and management of preeclampsia.

  • The manuscript is clear, well-structured, and well designed.
  • The study population and exclusion criteria are clearly defined.
  • The tables are clear and effectively illustrate the results.
  • The study’s limitations are appropriately acknowledged.

Overall, this is an interesting study that underscores the value of IPF. As the authors note, multicenter studies with larger cohorts are needed to further confirm the clinical utility of IPF.

Author Response

Response

We sincerely thank the reviewer for the positive and encouraging assessment of our work. We appreciate the reviewer’s recognition of the study design, clarity of presentation, and the clinical relevance of evaluating the immature platelet fraction in preeclampsia. We also value the acknowledgment of the study’s limitations and the emphasis on the need for larger multicenter studies, which aligns with our perspective and is noted in the manuscript.

Reviewer 2 Report

Comments and Suggestions for Authors

The study attempted to touch an important and relatively poorly studied feature of preeclampsia. The work appears to be carefully done, and the manuscript is well written. However, I have a few concerns which should be addressed before considering the paper for publication.

Major: the rationale of the study is not clear:

  • The association of preeclampsia and increased thrombogenic activity is well known. As I understand, although the study is stated as prospective, all analyses have been done at the time of diagnosis, i.e. at the late gestational age, and only once (if do not take into account post-treatment). What does the “diagnostic performance” mean if the diagnosis of PE has already been confirmed? What is the clinical significance of these findings, especially in respect to early diagnosis and therapeutic monitoring?
  • What did the authors mean as the dynamic insight? There are no data on the IPF indices obtained at different stages of pregnancy (at least during 2 trimester) which is more important for PE management. In the absence of such data, the conclusions are quite speculative. The diagnostic value is generally means an importance for early prediction. The paper simply shows the association between PE and platelet indices at the time of PE onset. Indeed, these data are interesting and important, but I would recommend to change the Title and the major conclusions (like IPF as a highly accurate and clinically relevant biomarker/predictor for preeclampsia).

Minor:  

Abstract:

  • Line 18: “post-treatment” – the type of treatment should be indicated here and in Materials and Methods.
  • Gestational age should be provided.

Results:

  • Why was the preeclampsia group not divided into early/preterm forms (requiring the delivery at < 34-37 weeks) and late-onset/term type? It should be done if possible.
  • Again, the rationale of determining the hematological changes after treatment is not clear and should be explained. I suppose these data are hardly new findings.

Author Response

Comment 1: The association of preeclampsia and increased thrombogenic activity is well known. As I understand, although the study is stated as prospective, all analyses have been done at the time of diagnosis, i.e. at the late gestational age, and only once (if do not take into account post-treatment). What does the ‘diagnostic performance’ mean if the diagnosis of PE has already been confirmed? What is the clinical significance of these findings, especially regarding early diagnosis and therapeutic monitoring?

Response 1: We sincerely thank the reviewer for this important and thoughtful comment. We agree that our study was not designed to evaluate the immature platelet fraction (IPF) as a predictive biomarker prior to the clinical onset of preeclampsia, and that clarification of the term “diagnostic performance” and the clinical relevance of our findings was warranted.

In this manuscript, the term “diagnostic performance” is used in its conventional analytical sense to describe the ability of IPF to discriminate women with clinically established preeclampsia from normotensive pregnant controls at the time of presentation, rather than to predict disease before diagnosis. Although preeclampsia had already been diagnosed according to ACOG criteria, our aim was to investigate whether IPF provides additional hematologic insight beyond routine platelet indices at this initial clinical evaluation.

Notably, IPF was markedly elevated at diagnosis, whereas platelet counts frequently remained within the normal range. This differential behavior suggests that IPF may reflect more sensitive alterations in thrombopoietic activity than platelet count alone, without implying a role in preclinical screening or early prediction.

Regarding clinical significance, we believe our findings may be relevant in two complementary contexts. First, at presentation, IPF may support diagnostic evaluation in situations where clinical features are evolving or borderline by providing additional hematologic context. Second, the significant early decline in IPF following treatment—despite stable platelet counts—suggests that IPF may reflect an early hematologic response to therapy. We present this observation cautiously as exploratory, as this pattern has not been well characterized in previous studies.

In response to the reviewer’s comment, we have revised the Discussion to clarify the intended interpretation of “diagnostic performance”, to avoid overstatement regarding early diagnosis or prediction, and to emphasize the potential role of IPF in diagnostic support and subtype-specific clinical monitoring rather than screening. The revised text is highlighted in red in the manuscript.

Below are some revised paragraphs (includıng the additions shown in red in the text) as examples for clarity:

In this study, although the diagnosis of preeclampsia was established according to ACOG criteria, IPF was evaluated to provide additional hematologic insight beyond clinical confirmation….

At the time of diagnosis, IPF was markedly elevated, whereas platelet counts frequently remained within the normal range, suggesting that IPF may reflect more sensitive alterations in thrombopoietic activity than conventional platelet indices…..

Importantly, the significant early decline in IPF following treatment—despite stable platelet counts—indicates that IPF may capture an early hematologic response to therapy….

We believe that these revisions address the reviewer’s concerns and more accurately align the interpretation of our findings with the scope and design of the study.

Comment 2: What did the authors mean as the dynamic insight? There are no data on the IPF indices obtained at different stages of pregnancy (at least during 2 trimester) which is more important for PE management. In the absence of such data, the conclusions are quite speculative. The diagnostic value is generally means an importance for early prediction. The paper simply shows the association between PE and platelet indices at the time of PE onset. Indeed, these data are interesting and important, but I would recommend to change the Title and the major conclusions (like IPF as a highly accurate and clinically relevant biomarker/predictor for preeclampsia).

Response 2:

We thank the reviewer for this important and constructive comment. We agree that the term “dynamic insight” may have been misleading, as our study did not include longitudinal IPF measurements across different stages of pregnancy and was not designed to evaluate IPF as an early predictive marker.

In response to this concern, we have removed the term “dynamic” throughout the manuscript and revised the wording to refer more precisely to early post-treatment hematologic changes observed after clinical diagnosis, rather than gestational-stage trends. We have also revised the title and major conclusions to avoid overstatement and to clearly reflect that our findings demonstrate associations between preeclampsia and platelet indices at disease onset, rather than predictive or longitudinal patterns.

Furthermore, we have clarified in the Discussion that the diagnostic performance of IPF in this study should not be interpreted as evidence for early prediction. Instead, it is presented as diagnostic discrimination at presentation and as supportive information for clinical assessment and monitoring. We have also clarified in the Methods section that post-treatment measurements were included to characterize short-term hematologic responses following routine management, rather than to imply longitudinal assessment.

For transparency, examples of the revised text (highlighted in red in the manuscript) are provided below. We believe these revisions address the reviewer’s concerns and more accurately align the manuscript with the scope and design of the study.

 Examples of revised text added to the manuscript and colored in red:

Discussion:

In this study, although the diagnosis of preeclampsia was established according to ACOG criteria, IPF was evaluated to provide additional hematologic insight beyond clinical confirmation. These findings are presented cautiously as exploratory and should not be interpreted as evidence for early prediction. Rather, they suggest a potential role for IPF in supporting diagnostic assessment at presentation and in subtype-specific clinical monitoring.

Materials and Methods:

The difference between these two measurements was calculated and reported as the delta (Δ) value, reflecting short-term post-treatment change.

Title (revised):

The Thrombopoietic Signature of Preeclampsia: Diagnostic and Monitoring Insights from the Immature Platelet Fraction

Comment 3:

Abstract: Line 18: “post-treatment” – the type of treatment should be indicated here and in Materials and Methods. Gestational age should be provided.

Response 3:

We thank the reviewer for this important clarification. We agree that specifying the post-treatment period and gestational age improves the transparency and clinical interpretability of the study. Accordingly, we have revised the Abstract and Materials and Methods sections to clarify that follow-up measurements were obtained 24–48 hours after initiation of medical treatment or delivery. We have also specified that gestational age. These revisions provide a clearer definition of the clinical time frame and fully address the reviewer’s concern. The revised text is highlighted in red in the manuscript.

Comment  4:

Results : Why was the preeclampsia group not divided into early/preterm forms (requiring the delivery at < 34-37 weeks) and late-onset/term type? It should be done if possible.

Response 4:

We thank the reviewer for this valuable suggestion. We agree that classification of preeclampsia according to gestational age at onset provides important biological and clinical insight and may help to elucidate subtype-specific hematologic patterns.

Accordingly, we have revised the manuscript to incorporate gestational age–based subgroup analyses. As described in the Materials and Methods section, preeclampsia was classified as early-onset (<34+0 weeks) or late-onset (≥34+0 weeks), reflecting established concepts of biological heterogeneity, with early-onset disease primarily related to impaired placentation and late-onset preeclampsia more closely associated with maternal cardiovascular, metabolic, and inflammatory factors.

In the Results section, we have added a new subsection entitled “Gestational Age–Based Differences in Δ Laboratory Parameters” (Section 3.5) and included Table 4, which presents comparative analyses of post-treatment changes in hematologic parameters between early- and late-onset preeclampsia. These analyses revealed subtype-specific differences, particularly with respect to ΔIPF.

In addition, we have expanded the Discussion to address the clinical and biological implications of these gestational age–related differences and to contextualize our findings within the existing literature on early- versus late-onset preeclampsia.

The revised text and newly added sections are highlighted in red in the manuscript.

Example of revised text added to the manuscript and colored in red:

…Independently of severity, preeclampsia is also classified according to gestational age at onset as early-onset (<34+0 weeks) or late-onset (≥34+0 weeks), reflecting its biological heterogeneity, with early-onset disease primarily related to impaired placentation and late-onset preeclampsia more closely associated with maternal cardiovascular, metabolic, and inflammatory factors [14].; added in materials methods section.

Comment 5:

 Again, the rationale of determining the hematological changes after treatment is not clear and should be explained. I suppose these data are hardly new findings.

Response 5: We thank the reviewer for this important comment. We agree that the post-treatment findings should not be overinterpreted as novel or definitive, and we have revised the manuscript accordingly to avoid any overstatement regarding originality.

The rationale for including post-treatment measurements was not to claim novelty, but to explore whether IPF behaves differently from conventional platelet indices during the early clinical course following routine management. In clinical practice, platelet counts often remain stable in the early post-diagnostic period despite therapeutic intervention, and we therefore aimed to assess whether IPF may capture early hematologic responses to treatment that are not reflected by platelet count alone. In response to the reviewer’s concern, we have removed statements implying that this was the first study to report post-treatment IPF changes and have revized the Methods and Discussion sections, we have reorganized these analyses as short-term hematological responses after treatment, rather than suggesting novelty or longitudinal evaluation.

We believe this clarification better defines the scope of the post-treatment analyses and addresses the reviewer’s concern regarding their interpretation. The revised text is highlighted in red in the manuscript.

Round 2

Reviewer 2 Report

Comments and Suggestions for Authors

Table 2, PLT-1: please, check p < 0.00*.

Table 4: since both early- and late-onset PE can be non-severe and severe, please, indicate that the parameters have been obtained in total patient population.

Author Response

Comment 1: Table 2, PLT-1: please, check p < 0.00*.

Response:

We thank the reviewer for this comment. The p-value in Table 2 (PLT-1) was rechecked and corrected to <0.001 in the revised table.

Comment 2: Table 4: since both early- and late-onset PE can be non-severe and severe, please, indicate that the parameters have been obtained in total patient population.

Response:

We thank the reviewer for this important clarification. To address this point, we revised the section heading, table title, and the opening sentence of the Results section to explicitly indicate that gestational age–based comparisons were performed in the total preeclampsia population, irrespective of disease severity. These revisions were made to avoid any potential ambiguity regarding patient subgroups.